# Assessment of a PCL-3D Printing-Dental Pulp Stem Cells Triplet for Bone Engineering: An In Vitro Study

**DOI:** 10.3390/polym13071154

**Published:** 2021-04-04

**Authors:** Raúl Rosales-Ibáñez, Nieves Cubo-Mateo, Amairany Rodríguez-Navarrete, Arely M. González-González, Tomás E. Villamar-Duque, Leticia O. Flores-Sánchez, Luis M. Rodríguez-Lorenzo

**Affiliations:** 1Tissue Engineering Lab, Facultad de Estudios Superiores Iztacala, Universidad Nacional Autónoma de México (UNAM), Av. Tenayuca-Chalmita S/N, Cuautepec Barrio Bajo, Alcaldía Gustavo A. Madero, Ciudad de México CP. 07239, Mexico; cd.anyrn@outlook.com (A.R.-N.); Arely_Glezz@hotmail.com (A.M.G.-G.); 2Sensors and Ultrasonic Systems Department, Institute for Physical and Information Technologies, (ITEFI-CSIC), C/ Serrano 144, 28006 Madrid, Spain; n.cubo.mateo@gmail.com; 3Universidad Internacional de Valencia, C/ Pintor Sorolla 21, 46002 Valencia, Spain; 4Bioterio, Facultad de Estudios Superiores Iztacala, Universidad Nacional Autónoma de México (UNAM), Av. De los Barrios No.1, Tlalnepantla, Estado de México CP. 54090, Mexico; vidutoer@yahoo.com.mx (T.E.V.-D.); letyflores1107@gmail.com (L.O.F.-S.); 5Department of Polymeric Nanomaterials and Biomaterials, Institute Science and Technology of Polymers (ICTP-CSIC), C/ Juan de la Cierva 3, 28006 Madrid, Spain

**Keywords:** polycaprolactone, dental pulp stem cells (DPSCs), 3D printing, scaffolds, bone regeneration

## Abstract

The search of suitable combinations of stem cells, biomaterials and scaffolds manufacturing methods have become a major focus of research for bone engineering. The aim of this study was to test the potential of dental pulp stem cells to attach, proliferate, mineralize and differentiate on 3D printed polycaprolactone (PCL) scaffolds. A 100% pure M_w_: 84,500 ± 1000 PCL was selected. 5 × 10 × 5 mm^3^ parallelepiped scaffolds were designed as a wood-pilled structure composed of 20 layers of 250 μm in height, in a non-alternate order ([0,0,0,90,90,90°]). 3D printing was made at 170 °C. Swine dental pulp stem cells (DPSCs) were extracted from lower lateral incisors of swine and cultivated until the cells reached 80% confluence. The third passage was used for seeding on the scaffolds. Phenotype of cells was determined by flow Cytometry**.** Live and dead, Alamar blue™, von Kossa and alizarin red staining assays were performed. Scaffolds with 290 + 30 μm strand diameter, 938 ± 80 μm pores in the axial direction and 689 ± 13 μm pores in the lateral direction were manufactured. Together, cell viability tests, von Kossa and Alizarin red staining indicate the ability of the printed scaffolds to support DPSCs attachment, proliferation and enable differentiation followed by mineralization. The selected material-processing technique-cell line (PCL-3D printing-DPSCs) triplet can be though to be used for further modelling and preclinical experiments in bone engineering studies.

## 1. Introduction

Tissue engineering (TE) has been defined as the science of persuading the body to regenerate or repair tissues that fail to regenerate or heal spontaneously [1]. The main approach in tissue engineering utilizes cells, signals and scaffolds for the design and construction of biological substitutes that should result in tissue regeneration [2]. Biomaterials are used as scaffolds for spatial distribution of specific cell types [3]. Biomaterials can potentially influence cell proliferation and extracellular matrix formation, both in positive and negative ways [4]. Also, the selection of the material determines the most appropriate manufacturing technique for the scaffold. Scaffolds should mimic as close as possible the features of the extracellular matrix of the tissue to be rebuilt, therefore allowing colonization, proliferation and functional differentiation of cells within the scaffold [5]. General requirements for scaffolds are: ease of handling, adequate porosity, adequate biodegradability rate and non-toxic degradation products, good physical and mechanical resistance, low immunogenicity and the ability to host vascularization [6,7].

Polycaprolactone (PCL) is one of the most extensively studied biodegradable polyesters for bone engineering [8]. The degradation product of PCL is 6-hydroxyhexanoic acid, which is naturally metabolized by the human body. The toxicology of polycaprolactone devices have been studied and mentioned as safe devices for several clinical treatments [9]. PCL favorable properties for bone engineering include long-term biodegradability, ease of processing due to the low melting temperature (58–60 °C), suitable rheological, viscoelastic properties and thermal stability. Because of its history in drug delivery devices, PCL has also a shorter regulatory path to market than many other polymer systems through the Food and Drug Administration (FDA) and the European Medicines Agency (EMA) [10]

The production of scaffold suitable for tissue engineering has been approached using different techniques. Among them, supercritical drying [11] and 3D printing [10] are the most innovative. Additive manufacturing (AM) techniques are becoming the techniques of choice for the development of scaffolds as implants and in tissue engineering. Their main advantage is the ability of designing the geometrical parameters of the scaffolds in advance, and the potential to tailor and scale the manufacturing of scaffolds to each experimental model or patient. In addition, 3D printing has significantly increased the economic feasibility of low volume production runs, because for the majority of traditional manufacturing methods, like injection molding, costs can only be recouped for high volume production runs [12]. Three-dimensional printed medical grade polycaprolactone scaffolds were introduced more than 10 years ago as a scaffold for bone tissue engineering [13] and tested in complicated craniomaxillofacial defects showing no signs of infection and fusion with existing native tissue. Results attributed to early vascularization and tissue filling within the pores [14].

Different sources of stem cells have been assayed for different applications in tissue engineering, they include bone marrow adipose-derived and bone marrow stem cells for maxillary bone defects [15], placenta have shown the potential to support bovine mesenchymal stem cell adherence and differentiation into osteogenic and chondrogenic lineages [16], umbilical cord blood have been used to obtain hematopoietic stem and progenitor cells for the treatment of hematological disorders and as a supportive therapy for malignant diseases [17], menstrual blood-derived endometrial stem cells have proven to be effective in the treatment of severe Asherman syndrome [18], adipose tissue stem cells have been used in the development of constructs for bone regeneration [19]. The interest in the usage of dental stem cells for tissue engineering or cell-based therapies is increasing due to their accessibility, because they proliferate at a fast rate and they are able to differentiate in vitro or in vivo into functional odontoblasts [20]. Potential applications for these cells have been described not only for oral applications but also for orthopedic and maxillofacial reconstruction [21]. Periodontal ligament stem cells (PLSCs) have been used for the regeneration of bone defects caused by periodontal disease both in animal models [22] and in clinical practice [23,24] and dental pulp stem cells as cell therapy for autoimmune and related disorders [25]. Swine dental pulp stem cells (DPSCs) are similar to their human counterparts exhibiting mesenchymal stem cell characteristics with ability to form colony forming unit-fibroblastic and odontogenic differentiation potential [26] and therefore useful for creating constructs in large animal model studies [27].

Studies covering high throughput screening of over 1700 embrionic stem cell-materials interactions have been published [28], however it have also been shown that for the same biomaterial, different (stem) cells can display discrepancies in terms of cell adhesion, morphology, viability, proliferation, and apoptosis due to cell type specific and species-specific differences [29] and for particular applications such as, for example, endodontic regenerative approaches the size and the density of pores must be perfectly controlled for allowing cell migration and proliferation, transport of nutriments and active molecules and elimination of wastes [30], thus a holistic approach including the cell type, the biomaterial selected and the scaffold properties seems necessary for each particular application.

The aim of this study was to test the potential of dental pulp stem cells to attach, proliferate mineralize and differentiate on 3D printed polycaprolactone scaffolds in order to set up a first step to develop a therapeutic regenerative model for oral applications.

## 2. Materials and Methods

### 2.1. Scaffolds Manufacturing

The polycaprolactone (PCL) used in this project was directly provided as a 1.75 ± 0.005 mm filament for standard 3D printers (3D4makers.com (accessed on 7 March 2021), M_w_: 84,500 ± 1000, 100% pure, Haarlem, The Netherlands). Polymer characterization is described somewhere else [10].

Scaffolds manufacturing was made with a Hephestos 2 (Prusa i3-BQ, Madrid, Spain) equipped with a heated bed working at 40 °C. The extruder was a double drive gear, with a nozzle of 400 μm, set at 170 °C. Cooling down was forced with an external fan.

Each parallelepiped scaffold (5 × 10 × 5 mm^3^) was designed as a wood-pilled structure composed of 20 layers of 250 μm in height, in a non-alternate order ([0,0,0,90,90,90]). The scaffolds were printed with no external shell, but with a closed thin layer in the bottom continued in the borders as a 1 mm brim. Images of the manufactured scaffolds and pore morphology were obtained with an optical microscope (Y-FL eclipse 400, Nikon, Tokyo, Japan). Pore morphology was determined by imaging cut samples using a XL30, scanning electron microscope (Phillips, Eindhoven, The Netherlands), operating at 25 kV. Specimens were coated with Au-Pd using a Polaron SC7640 (Quorum Tech. LTD, Kent, UK) sputter coater. Fiji software (ImageJ v1.52 h, U.S. National Institutes of Health, Bethesda, MD, USA) was used for image analysis [31].

### 2.2. Swine Pulp Stem Cells (sDPSCs) Isolation and Culture

Six months old female four Vietnamese swine were maintained in laboratory conditions in the “Bioterio” of the Superior Studies Faculty at Iztacala (FESI-UNAM), in accord with Mexican normativity (NOM-062-ZOO-1999). Animal care and use were approved by the ethical committee (CE/FESI/052017/1174). Lower lateral incisors of each swine were extracted to isolate the stem cells. The pulp was cut into small pieces and treated with 0.05% collagenase (Sigma Aldrich, St. Louis, MI, USA) and low glucose Dulbecco’s modified Eagle Medium (DMEM) (Bio-west, México) and incubated at 37 °C for 20 min. After the collagenase inactivation, the cell suspension was centrifuged at 350 g for 5 min, the cell pellet plated in T25 tissue culture flask (TPP-Techno Plastic Products, Trasadingen, Switzerland) with low glucose Dulbecco’s modified Eagle Medium (DMEM) (Bio-west, México), supplemented with 10% Fetal Calf Serum (FCS) (Bio-west) and 1% penicillin–streptomycin (Sigma Aldrich), and incubated at 37 °C in an atmosphere of 5% CO_2_. After three days in culture, the non-adherent hematopoietic cells are washed away. The culture media was changed twice a week until the cells reached 80% confluence. The third passage was used for the following tests.

### 2.3. Phenotype of Swine Dental Pulp Stem Cells by Flow Cytometry

Expression of surface markers of stem cells was analyzed by flow cytometry using monoclonal antibodies conjugated CD90–FITC, CD73–PECY7, CD105–VB421 (BD, Franklin Lakes, NJ, USA). The cells were suspended and incubated in phosphate buffered saline (PBS) (Bio-west, México) supplemented with 2% Fetal Calf Serum (FCS) (Bio-west) and containing monoclonal antibodies in a 1:1000 solution for 20 min at room temperature. Analysis was performed on CytoFLEX LX (BRVYNI, Beckman Coulter, Brea, CA, USA). The data were processed using FlowJo software (FlowJo, LLC, Ashland, OR, USA).

### 2.4. Live and Dead Assay

Cell viability was assessed with a live/dead assay (Live/Dead, Invitrogen^TM^, Carlsbad, CA, USA) according to the manufacturer’s protocol. In this case, calcein AM (0.5 µL) and ethidium homodimer-1 (2.0 µL) were dissolved in 997.5 µL PBS, added to the samples and incubated for 30 min while protected from light at 37 °C in a humidified atmosphere with 5 % CO_2_. The samples were evaluated with a 40x objective epifluorescence microscope with a ZEISS SCOPE.A1microscope (Software ZEN LIET 2012, Carl Zeiss AG, Oberkochen, Germany).

### 2.5. Alamar Blue™ Assay

Cell viability was determined using the colorimetric Alamar blue™ (Sigma-Aldrich) cell metabolic assay. The manufactured PCL scaffolds were deposited in 96 well culture plates. Cells (8 × 10^6^) were seeded on the scaffolds or added directly on 96-well culture plates (control group), In both cases, cells were cultured with DMEM, 10% bovine serum, and 1% penicillin -streptomycin. Incubation conditions included 37 °C, 5% CO_2_ atmosphere and 95% humidity. After 3, 7 and 10 days, the medium was removed, we added 90 µL of fresh medium and 10 µL of Alamar blue™, and the cells were incubated for 4 h under the same mentioned conditions. Cell viability was read at 540 nm with a microplate reader (Biotek Elx808, Winooski, VT, USA). Three repetitions were performed for each experimental condition. The samples were evaluated using a microplate reader.

### 2.6. Osteogenic Differentiation of Swine Dental Pulp Stem Cells in a 3D Scaffold

A cell density of 2 × 10^6^ of swine DPSCs were seeded on a 96-well plate (group A). A cell density of 1 × 10^6^ of swine DPSCs were seeded on one of the surfaces of the 3D scaf-folds (group B and C). After 5 min, the scaffolds were turned over and a cell density of 1 × 10^6^ of cells were seeded on the opposite surface of the scaffolds. The three groups were culture with low-glucose DMEM (Bio-west, México), supplemented with 10% Fetal Calf Serum (FCS) (Bio-west, México) and 1% penicillin–streptomycin (Sigma Aldrich), and incubated at 37 °C, 5% CO_2_ under humidification for 48 h. After that time, the culture medium was removed and replaced with new medium for groups A and B. The culture medium was changed twice a week during the 28-day experimental period. The swine DPSCs seeded on the 3D scaffolds (group C) were cultured with MesenCult™ osteogenic differentiation medium (constituted with basal medium, 15 µL of dexame-thasone, 250 µL of ascorbic acid, 175 µL of β-glycerophosphate and 7.5 mL of Mesen-Cult™ osteogenic stimulatory supplement (Stem Cells Technologies, Cambridge, MA, USA), changed twice a week during the 28 days of induction. Subsequently, the following stains were made.

### 2.7. Von Kossa Staining

This method was used to visualize the formation of calcium phosphate particles in histological stains. The presence of calcium phosphate deposits is stained black. The principle of this coloration is based on the transformation of calcium salts into silver salts: calcium ions, bound to phosphates, are replaced by silver ions brought by a solution of silver nitrate. 

The cell monolayers were fixed with 4% neutral formalin (Sigma-Aldrich, USA) on 96-well plates and scaffolds. The NovaUltraTM von Kossa Stain Kit (ABCAM, Cambridge, UK) samples were incubated with Silver Nitrate Solution (5%) for 60 min with exposure to ultraviolet light, samples were washed three times with distilled water, and incubated with sodium. The samples were evaluated using a Leica DM IL LED (Leica Microsystems) inverted light-field phase contrast optical microscope.

### 2.8. Alizarin Red Staining (ARS)

Alizarin red S colored calcium deposits selectively and it has been used for decades to evaluate calcium-rich deposits in cell culture [32]. Well plates and 3D PCL scaffold specimens were seeded with swine DPSCs, fixed with 4% neutral formalin (Sigma-Aldrich, USA) for 5 min and then washed three times with PBS. The fixed cells over scaffolds were further washed with distillated water in order to remove any salt residues. A solution of 2% wt/v Alizarin Red S (Sigma Aldrich, USA), with adjusted a pH 4.2 was added, so that it covered the entire surface of the scaffolds. After 10 min of incubation at room temperature, the ARS excess was washed with distillated water. The ARS staining was imaged using a Leica DM IL LED inverted light-field phase contrast optical microscope. 

### 2.9. Statistical Analysis

Cell viability and proliferation results were expressed as the mean value ± standard deviation. Data were analyzed within experimental groups with ANOVA test and post-hoc Tukey’s honestly-significant-difference (HSD) test to evaluate temporary variations. Student’s *t* tests were performed to detect differences between control and scaffolds groups at each time. A *p* value < 0.001 was considered as significant.

## 3. Results

### 3.1. PCL Scaffolds

Scaffolds with 290 ± 30 μm strand diameter, 938 ± 80 μm pores in the axial direction and 689 ± 13 μm pores in the lateral direction were obtained as it has been calculated from Figure 1. These data correspond to a 1.37 ± 0.22 μm axial to lateral ratio. The fast cooling down yielded faster filament setting, generating a cylindrical strand that resulted in a regular, open-pore structure avoiding pore collapsing and generating a more open scaffold with high shape fidelity.

### 3.2. Cell Extraction and Cultivation

Swine dental pulp stem cells were cultivated from lower lateral incisor from four specimens by means of enzymatic digestion [33,34]. On the second day, the culture medium was removed, since there were many non-adherent hematopoietic cells. On days 4 to 6 the first cell with fusiform shape fibroblast-like cells were observed, on day 21 a 70–80% confluence was reached as shown in the Figure 2A,B.

### 3.3. Phenotype of Swine Dental Pulp Stem Cells by Cytometry Analysis

The identification of multi-labeling marker panels was used for the identification of CD73, CD90 and CD105 of the cells obtained from swine dental pulp. The results collected from Figure 3 indicated that they were 96.04%, positive for both CD90 and CD105, 97.74% positive for both CD90 and 95.37% positive for CD73 and CD105. 

### 3.4. Live and Adhesion Cells Assay on 3D Printed PCL Scaffolds

In order to differentiate live/dead and adhesion cells from swine DPSCs on the 3D printed PCL scaffolds, we used the live/dead cell assay, in which green fluorescence is observed in live cells, while dead cells show bright red fluorescence. Figure 4A,B shows the epi fluorescence microscopy images of the control covers round objects after 3 incubation days (A) and control covers round objects after 7 incubation days (B). Figure 4C,D shows the epifluorescence microscope images of swine DPSCs seeded on 3D PCL scaffold after 3 incubation day (C) and swine DPSCs seeded on 3D PCL scaffold after 7 incubation days (D). The separation of red and green channels enables to appreciate a higher proportion of live than dead cell on 3D PCL printing scaffolds than on the covers round objects on 24-well plates used as control.

### 3.5. Alamar Blue™ Viability Cells on 3D Printed PCL Scaffolds

Cell viability results are shown in Figure 5. The cell viability increased significantly through time both in control (F_(2,6)_ = 66.50; *p* < 0.001), and in scaffolds groups (F_(2,6)_ = 506.75; *p* < 0.001). The post-hoc Tukey HSD test confirmed significant differences in control group between data at day 3 (0.22 ± 0.001), day 7 (0.35 ± 0.001), and day 10 (0.60 ± 0.001). In the PCL scaffolds, correspondent results (0.20 ± 0.001); (0.31 ± 0.004); and (0.58 ± 0.026), were significantly different as well (*p* < 0.001 in all cases). Student’s *t* tests demonstrated that cell viability was significantly higher in control group than in PCL scaffolds at days 3 (*t* = 33.98; *p* < 0.001) and 7 (*t* = 16.70; *p* < 0.001) but was similar at day 10 (*t* = 1.52; *p* = 0.202).

### 3.6. Osteogenic Differentiation and Mineralization on PCL 3D Scaffolds Seeded with Swine DPSCs

Swine DPSCs seeded in PCL 3D printing scaffold were surrounded by an osteogenic matrix-like substance where calcium phosphate particles appeared as black regions in von Kossa, Figure 6 von Kossa staining can be used as indicator of osteogenic differentiation. according to Declercq et al. [35]. A higher calcium deposition can be observed on the DPSCs seeded on 3D PCL printing + MesenCult™ Osteogenic Differentiation Kit, than on the DPSCs seeded in the 3D PCL printing scaffold using only DMEM as medium but also a higher calcium deposition is observed in this one than on the control group. As described in the literature, during early osteogenic induction, stem cells continue to proliferate and migrate [36]. Some cells have adhered to the scaffolds, but others cells have migrated from the scaffold to the bottom of the well plates or to the surface of the flask, as it can be observed in Figure 6A,B [37]. Then, some of these cells have begun to secrete extracellular matrix at the bottom of the well-plate and also on the scaffolds as it can be appreciated in Figure 6C. However, a greater density can be appreciated on the scaffolds than on the well plate. 

Also, photographs of the Alizarin red staining, Figure 7, show higher calcium deposition on the DPSCs seeded on 3D PCL printing + MesenCult™ Osteogenic Differentiation Kit than on the DPSCs seed on the 3D PCL printing scaffold cultured only with DMEM.

## 4. Discussion

Additive manufacturing techniques has recently gained popularity for tissue engineering studies because these material processing techniques allow for building scaffolds with highly regular morphology and completely interconnected pores and channels, and with the advantage that they can be designed in advance [38]. Figure 1 enables the appreciation of the regular morphology and interconnectivity of the macropores achieved on the printed scaffolds. Such morphologies or/and interconnectivity of the pores can be achieved with some other processing techniques, such as supercritical drying [11], freeze-drying [39], particulate or foam leaching [40], electrospinning [41], emulsion template [42] starch consolidation [43] or cryopolymerization [19,44,45] but it is difficult to design them in advance. However, these techniques present specific advantages such as enhanced cell adhesion or generating fibrillar scaffolds that mimic the structure of the Extra Cellular Matrix (ECM) in the case of electrospinning [41], freeze drying is a good choice for processing polysaccharides [46], particulate or foam leaching for combination of bioactive and resorbable components, thus, for developing a porous structure “in situ” that enable the ingrowth of new born tissue [40], emulsion template have produced degradable porous scaffolds suitable for the 3D culture of keratinocytes and fibroblasts [42], starch consolidation have been used for the production of ceramic porous scaffolds [43], cryopolymerization have shown to be useful for introducing chemical and physical cues that enhance osteogenesis of bone marrow mesenchymal stem cells in bone regeneration [45] and supercritical drying have been used to obtain aerogels that maintain their native morphology at micro and nanoscale and may be useful for mimicking blood vessels [11].

The ability to facilitate cell attachment and induce cell proliferation is one of the frequently mentioned concerns related with the use of additive manufacturing techniques [10]. Though, pore morphology obtained with 3D printing should facilitate cell proliferation [47], complementary techniques are proposed to overcome this problem and generate additional topography features than increment cell adhesion and guides for cell proliferation [48]. However, careful control of the printing parameters allows the manipulation of these features in 3D printing as shown in a previous work [10]. In the current manuscript we investigate the potential for DPSC attachment and proliferation of a 3D printed scaffold without introducing further topography features. 

As stated in the introduction, the establishment of suitable material-processing technique-cell line triplets is the key for developing tissue engineering approaches. Regenerative therapies may take two different approaches, cell seeded scaffolds [49], which require a prior ex vivo step where stem or differentiated cells are grown on the scaffold to prepare implantable constructs or cell homing where an “in situ” cell recruitment and colonization is required [50]. The second one relies more in the body ability to heal and it may have a shorter period for translation to the clinic [51] but both approaches depend on the capacity of the material [52] and manufactured device to induce or facilitate tissue ingrowth [53]. We are proposing the use of DPSCs in combination with 3D printed PCL scaffolds as the baseline for developing preclinical large animal models but also for the manufacturing of 3D models for in vitro studies that can substitute or reduce the need for animal models in the development and testing of new treatments in agreement with international animal care guidelines. In a 10-year review of orbital floor reconstructions in patients implanted, PCL was found to perform better clinically than other bioresorbable implants made from poly(glycolide) (PGA), poly(lactic-co-glycolic acid) or poly(lactic acid) [13]. PGA degraded within 4 months which is too quick for the regeneration process and the acid degradation products of PLA [54] caused complications such as sinus formation and osteolysis attributed fluid accumulation [55]. The success of PCL implants is attributed to the recreation by 3D printing of a biomimetic microstructure that mimics the trabecular bone microstructure and encouraged vascularization and cell–cell communications [13].

The first step of this study was the isolation and characterization of DPSCs from Vietnamese swine teeth. Isolated cells showed typical stem cell morphology, characteristic plastic adherence and phenotype, denoted by the positive response for CD73, CD90 and CD105 protein surface markers during the flow cytometry as it can be appreciated in Figure 2 and Figure 3. Thus, the intrinsic properties of PCL, the pores size and the structural integrity of the 3D printed PCL scaffolds, did not seem to negatively affect the cell viability, attachment and proliferation. 

The presence and interaction of the cells attached in the scaffolds was confirmed by the fluorescence microscopy in the live and dead assay collected in Figure 4. The Alamar blue™ assay results displayed in Figure 5 demonstrated that cell viability keep increasing through time in both control and 3D PCL scaffold groups and was similar between the groups after 10 cultivation days. Mineralization nodules were also observed with von Kossa staining after 28 days of cultivation, collected in Figure 6, where the observed number of these nodules is larger than in the control group. The addition of an osteogenic differentiation media also produces an extension of the mineralization staining area in comparison with the scaffolds with no further supplementation. Together, von Kossa staining indicates that DPSC can differentiate and then mineralize without additives on the 3D printed PCL scaffolds, thus, a very light positive effect, but an effect anyhow, is suggested by this experiment. It also suggests that supplements should play a role for accelerating this process. This effect was attempted to confirm with an alizarin red staining and again higher calcium deposition is observed on the scaffolds than on the control group and also the addition of a supplementary osteogenic differentiation media results in higher calcium deposition, Figure 7. Together, cell viability tests, von Kossa and Alizarin red staining indicate the ability of the printed scaffolds to support DPSC attachment, proliferation and enable differentiation followed by mineralization, thus the selected material-processing technique-cell line triplet can be though to be used for further modelling and preclinical experiments.

## 5. Conclusions

PCL scaffolds with controlled architecture were produced by 3D printing. The scaffolds had not cytotoxic effect evaluated with DPSCs. The scaffolds support DPSCs adhesion and proliferation and posse a light effect on the mineralization capacity. The selected material-processing technique-cell line (PCL-3D printing-DPSCs) triplet can be though to be used for further modelling and preclinical experiments in bone engineering studies.

## Figures and Tables

**Figure 1 polymers-13-01154-f001:**
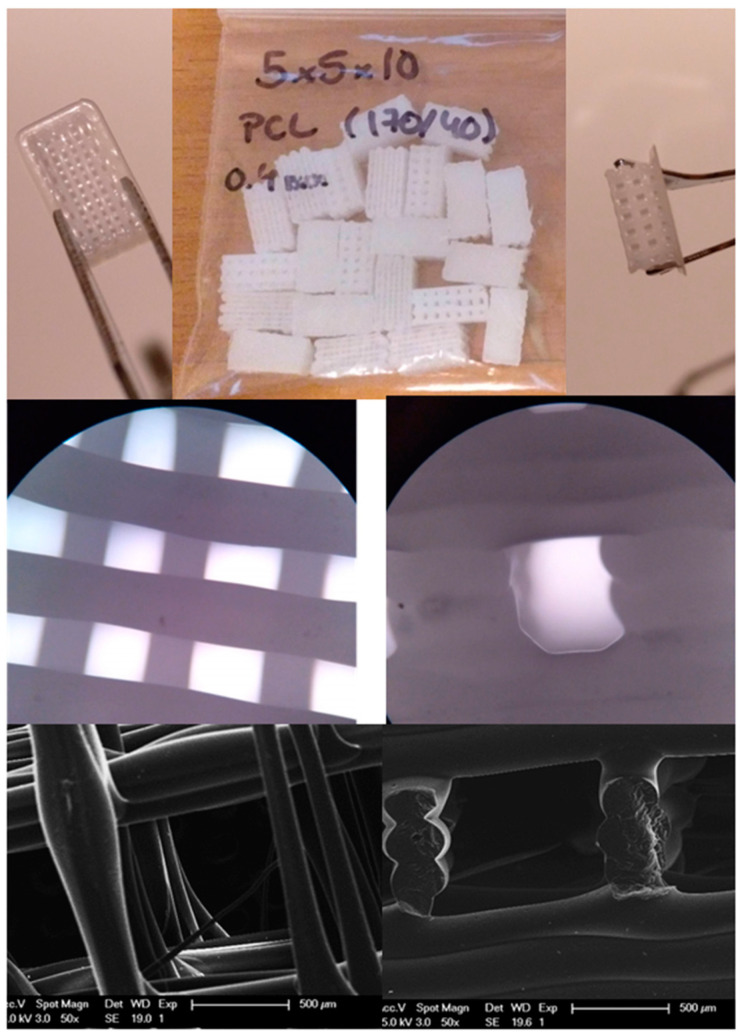
(**Top**). Centre, photograph of 3D printed scaffolds. (left), axial view and (right), lateral view of the scaffolds. (**Medium**). Axial optical microscopy (left) and lateral (right) views of the scaffolds. (**Bottom**). Scanning Electron Microscope (SEM) images from a top section (left) and from a lateral section (right).

**Figure 2 polymers-13-01154-f002:**
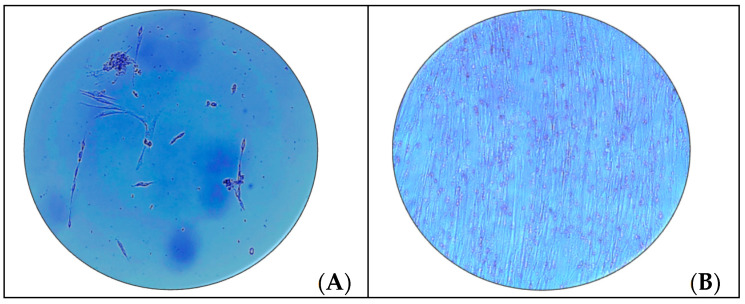
Morphology of swine Swine Dental Pulp Stem Cells (Swine DPSCs) on culture flask. On days 4 to 6 fusiform shape fibroblast-like cells were observed (**A**). On day 21 a 70–80% confluence was reached (**B**). Both images were taken in an inverted light-field phase contrast optical microscope at 10× magnification

**Figure 3 polymers-13-01154-f003:**
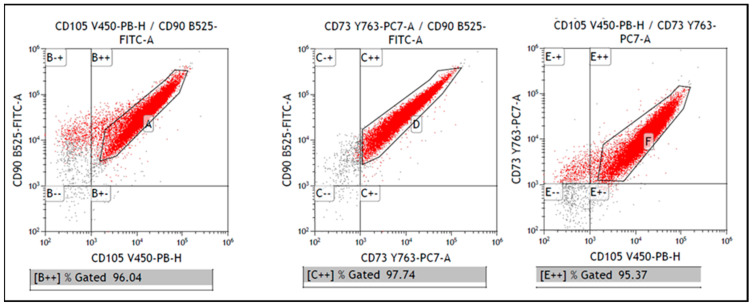
Representative bivariate dot plots. The phenotyping of the Mesenchymal Stem Cells (MSCs) by flow cytometry identified the expression of the surface proteins CD90 and CD105 in 96.04%, positive for both CD90 and CD73 in 97.74% and for CD73 and CD105 in 95.37%.

**Figure 4 polymers-13-01154-f004:**
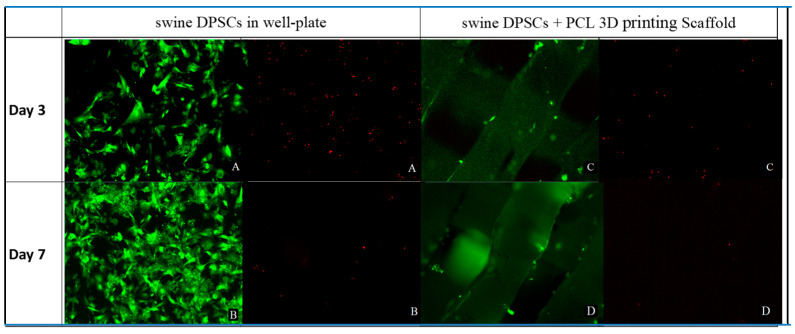
Cell viability and proliferation of swine DPSCs on the covers round objects on 24-well plates (**A**,**B**) and PCL 3D printed scaffolds (**C**,**D**). Cell viability and proliferation was determined by live/dead cell assay using calcein- AM (live in green) and ethidium homodimer-1 (dead in red). The results of the Live/Dead staining assay showed a higher proportion of live than dead cells on PCL 3D printed scaffolds than on the control. All images 40×.

**Figure 5 polymers-13-01154-f005:**
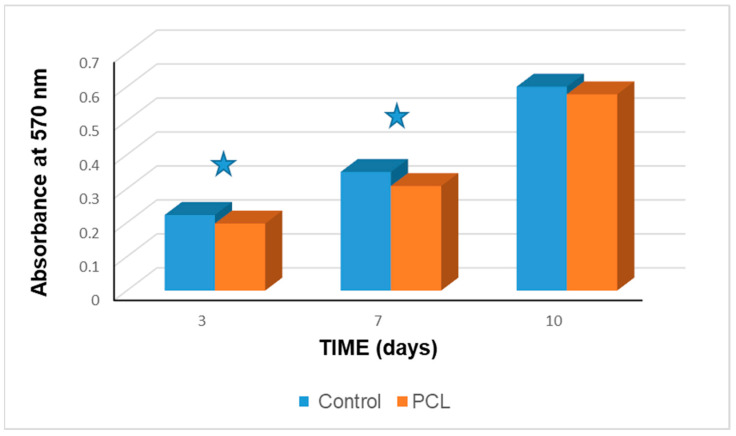
Cell viability (Alamar blue™ cell metabolic assay) in control and in 3D PCL scaffolds groups. Star symbol indicates statistically significant difference between groups (*p* < 0.01).

**Figure 6 polymers-13-01154-f006:**
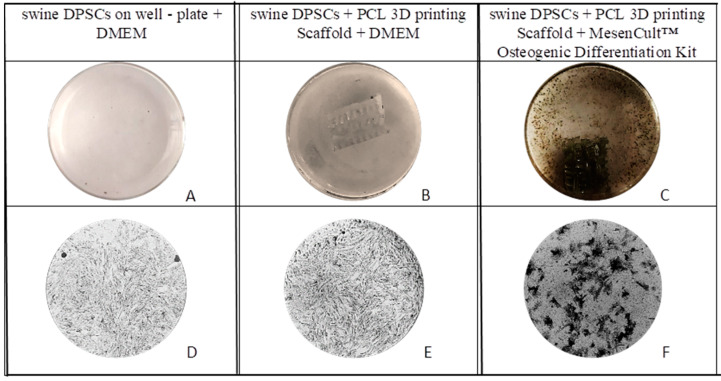
Representative images of the mineral deposition of swine Dental Pulp Stem Cells (DPSCs) on day 28 by von Kossa staining (top) (**A**–**C**), and respective microscopy images (**D**–**F**) at the bottom, 10×.

**Figure 7 polymers-13-01154-f007:**
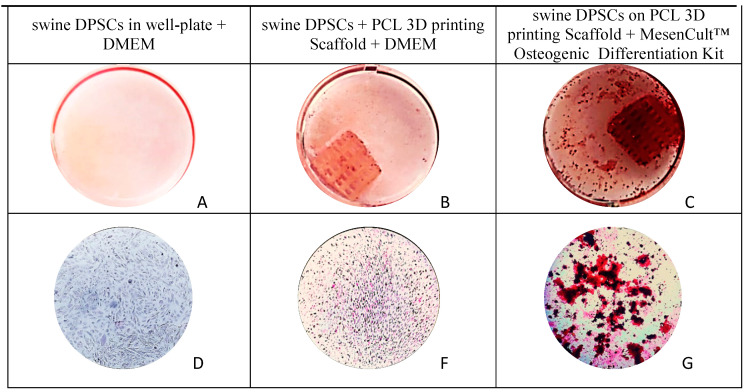
Representative images (top) of the mineral deposition in swine DPSCs on day 28 by Alizarin red S staining (**A**–**C**), and the respective microscopy images (**D**–**G**) at the bottom 10×.

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
