# Peer review of "Assessment of a PCL-3D Printing-Dental Pulp Stem Cells Triplet for Bone Engineering: An In Vitro Study"

_polymers, 2021, doi:10.3390/polym13071154_

Round 1
Reviewer 1 Report
The manuscript describes the fabrication of 3D printed PCL scaffold. Also, adhesion, proliferation, osteogenic differentiation and mineralization by Swine dental pulp stem cells within the scaffolds has been demonstrated. The cell viability has been assessed by means of live/ dead staining and Alamar blue assay. While, von Kossa and Alizarin Red S staining have been used to demonstrate the mineralization.
Overall, the paper presents a promising triplet for bone tissue engineering. The studies are well planned and systematically investigated.
Major comments:
- Line 232, 233- Data represented in Figure 4 is not sufficient to claim that the proportion of Live: dead is higher in case of 3D printed PCL scaffold as compared to the control. B column images are poor because of multiple planes available for the cells to grow. Confocal microscopy could fetch better images. Authors may also consider providing separate images for red channel.
- Line 276-279-The statement needs to be verified- the features namely, regular morphology, interconnected macropores can be achieved by other scaffold fabrication techniques such as cryopolymerization.
e.g. (a) Kemençe N, Bölgen N. Gelatin- and hydroxyapatite-based cryogels for bone tissue engineering: synthesis, characterization, in vitro and in vivo biocompatibility. J Tissue Eng Regen Med. 2017 Jan;11(1):20-33. doi: 10.1002/term.1813. Epub 2013 Aug 28. PMID: 23997022.
(b) Shalumon KT, Liao HT, Kuo CY, Wong CB, Li CJ, P A M, Chen JP. Rational design of gelatin/nanohydroxyapatite cryogel scaffolds for bone regeneration by introducing chemical and physical cues to enhance osteogenesis of bone marrow mesenchymal stem cells. Mater Sci Eng C Mater Biol Appl. 2019 Nov;104:109855. doi: 10.1016/j.msec.2019.109855. Epub 2019 Jun 5. PMID: 31500067.
- Figure 6- C It looks like the large number of DPSCs leave the scaffold and colonize the well plate. Shouldn’t majority of them remain adhered to the scaffold? Please clarify.
- Please elaborate the statement- ‘choosing a right combination of biomaterial, fabrication method and stem cell type is important’ with suitable examples.
- Please mention any background that formed the basis of scaffold design parameters.
Minor comments:
- Lines 25-26, lines 197-199 please fix the unit for scaffold dimensions
- Line 143 please specify if the same PCL scaffolds (5X10X5 mm3) were used in the Alamar Blue cell viability assay.
- Line 179, Please include the rationale of Alizarin Red S staining as mentioned in case of Alamar Blue assay or von Kossa staining.
- Figure 2- Authors are advised to provide the images with superior clarity.
- The manuscript requires professional language editing. Please pay attention to the correct usage of punctuation marks, choice of prepositions- e.g. line 207-208 should be “On the second day, the culture medium was removed, since there were many non-adherent hematopoietic cells.” Also, line 218-219 should be “The results collected on from figure 3 indicated..”
- Line 153-165- ‘Osteogenic differentiation of swine dental pulp stem cells in a 3D scaffold’ -please edit the paragraph to make it easier to understand for the readers.
- Lines 280-285 -revise these lines for the sentence construction is awkward, and meaning is not clear.
- Figure 3 legend- Representative bivariate dot plots. The phenotyping of the MSCs by flow cytometry. I believe it should be DPSCs.
- Line 220- CD73 is missing.
- Please comment on the biodegradability of the scaffolds and their ability to host vascularization
Reviewer 2 Report
- The title does not mention study design.
- Line 279 and 280. Sentence is not appropriate. Furthermore, advantages in many applications is not clear enough, please specify the applications.
- Line 71 and 72. Authors should also present periodontal ligament stem cells which is of dental origin and they should present any relevant study conducted on same topic with PDLSC.
- Line 78 and 79. Author should write specifically the applications of stem cells especially of oral origin in different therapeutic measures.
- Line 50. The authors should more clarify and justify the use of PCL scaffold by giving comparisons with other similar types of scaffolds.
- Overall: Nicely conducted study, however the authors did not clearly justify rationale for using dental pulp stem cells. It is suggested that authors should more clarify the use of dental pulp stems cells and not any other stem cell of dental origin.
Author Response
"Please see the attachment."

Reviewer 3 Report
The manuscript “Assessment of a PCL-3D printing-Dental Pulp Stem Cells triplet for bone engineering” deals with the production of biocompatible and biodegradable porous scaffolds seeded with stem cells, with the aim of developing a regenerative system for oral applications. The work showed interesting results from a biological and medical point of view; however, some revisions have to be performed.
In particular:
- Introduction. The production of scaffold suitable for tissue engineering has been approached using different techniques. Among them, 3D printing and supercritical drying are the most innovative. For this purpose, it should be interesting to cite also this last technique in the Introduction and in Discussion. Several works can be found in the literature, as, for example, the study of Baldino et al., A new tool to produce alginate-based aerogels for medical applications, by supercritical gel drying, Journal of Supercritical Fluids, 2019, 146, pp. 152-158; etc…
- Results. Please, empathize the relation and the effect of scaffold morphology, porosity and pore size on cell attachment, proliferation and migration.
- Some typing errors are present in the Abstract and in the text. Please, check and correct them.
- References are not in the format indicated by the Journal. Please, check and correct them.
Author Response
"Please see the attachment."

Round 2
Reviewer 1 Report
I am glad to see that the authors have satisfactorily addressed most of the queries. However, I am not quite satisfied with the cell proliferation data within the scaffold.
- In revised figure 4- day 3 for the selected plane of the scaffold, there are a substantial number of dead cells. Also, very few live cells are visible. Similarly, for day 7, too few live cells can be located on the scaffold. So, proportion-wise the number of dead cells is large.
- For the data represented in figure 5, with the superior architectural features such as regular morphology, interconnected pores, and 3D space available there should have been higher cell proliferation for the scaffold group as compared to the 2D control. It is seen that the cell density for control and scaffold becomes comparable by day 10. Monitoring the experiment further might have added some clarity.
In continuation with my earlier comment- “Figure 6- C It looks like the large number of DPSCs leave the scaffold and colonize the well plate. Shouldn’t the majority of them remain adhered to the scaffold?” I would like to suggest that scaffold modification to enhance the cell adhesion and proliferation (e.g. conjugation of RGD containing peptide) might improve the outcome. Authors are advised to touch upon this in the discussion section.
Reviewer 2 Report
Many thanks for the revision and incorporating all suggested changes to the manuscript
Reviewer 3 Report
The authors performed the modifications proposed by the Reviewer and improved the manuscript.